# Foldamers controlled by functional triamino acids: structural investigation of α/γ-hybrid oligopeptides
David Just [1], Vladimír Palivec[1], Kateřina Bártová[1], Lucie Bednárová[1], Markéta Pazderková[1], Ivana Císařová [2], Hector Martinez-Seara [1] ✉ & Ullrich Jahn [1] ✉

Peptide-like foldamers controlled by normal amide backbone hydrogen bonding have been extensively studied, and their folding patterns largely rely on configurational and conformational constraints induced by the steric properties of backbone substituents at appropriate positions. In contrast, opportunities to influence peptide secondary structure by functional groups forming individual hydrogen bond networks have not received much attention. Here, peptide-like foldamers consisting of alternating α,β,γ-triamino acids 3-amino-4-(aminomethyl)-2-methylpyrrolidine-3-carboxylate (AAMP) and natural amino acids glycine and alanine are reported, which were obtained by solution phase peptide synthesis. They form ordered secondary structures, which are dominated by a three-dimensional bridged triazaspiranoid-like hydrogen bond network involving the non-backbone amino groups, the backbone amide hydrogen bonds, and the relative configuration of the α,β,γ-triamino and α-amino acid building blocks. This additional stabilization leads to folding in both nonpolar organic as well as in aqueous environments. The three-dimensional arrangement of the individual foldamers is supported by X-ray crystallography, NMR spectroscopy, chiroptical methods, and molecular dynamics simulations.

The intricate interplay between biomacromolecules and their accurate three-dimensional organization are crucial for life. Proteins, one of three fundamental biopolymers composed of L-α-amino acids, fulfill their biological function by their defined arrangement, thus forming compact secondary and eventually tertiary structures. The most common secondary patterns are β-strands and α-helices ($3.6_{13}$-helix). Inspired by these evolutionary formed molecular architectures, artificial conformationally ordered systems termed foldamers got in the focus of research[1–6]. Among them, peptide-like oligomers incorporating non-natural amino acids have been widely studied because of their large structural diversity and functionalities resulting in potential applications in biomaterials, drug-delivery systems, and catalysis[7–14]. Several fundamental foldamer types have been investigated based on their basic building blocks, most prominently homooligomers assembled from α-[15–17], β-[18–22], or γ- amino acids[23–29] or hybrid oligomers constructed by their combination (Fig. 1a)[30–36]. The most studied non-natural foldamer class are β-peptides[18–22] having an extra carbon atom between the amino and carboxylate group compared to natural α-amino

acid peptides; a number of them were applied as important research tools and drug candidates[11–13,37]. Recently, γ-peptides[23–29] gained attention because the additional carbon atom opens new possibilities to tailor their conformational properties, which is commonly achieved by conformationally constraining the backbone by cyclic subunits. Three categories of constrained cyclic γ-amino acids have been applied in foldamers: the cycle is connected i) to the α- and β-carbon atoms ($\gamma^{2,3}$-peptides)[35,38,39], (ii) to the β- and γ-carbon atoms ($\gamma^{3,4}$-peptides)[31,32,40,41], or (iii) bridging the α- and γ-carbon atoms ($\gamma^{2,4}$-peptides)[42].

Research on foldamers concentrated mostly on the static relationship between given conformational constraints and hydrogen bonding between acceptors and donors along the backbone amide units in rigid structures (Fig. 1b), resulting in the identification of several helical folding patterns, such as 8-, 10-, 11-, 12/10-, 14-, or 14/15-helices[26]. Since it has so far not been possible to securely predict the formation of secondary structures, the conformational constraint, the stereochemical arrangement of the backbone, and intramolecular hydrogen bonding must be taken into account for

[1]Institute of Organic Chemistry and Biochemistry, Czech Academy of Sciences, Flemingovo náměstí 2, 16610 Prague 6, Czech Republic. [2]Department of Inorganic Chemistry, Faculty of Science, Charles University in Prague, Hlavova 2030/8, 12843 Prague 2, Czech Republic. ✉e-mail: hseara@gmail.com; ullrich.jahn@uochb.cas.cz

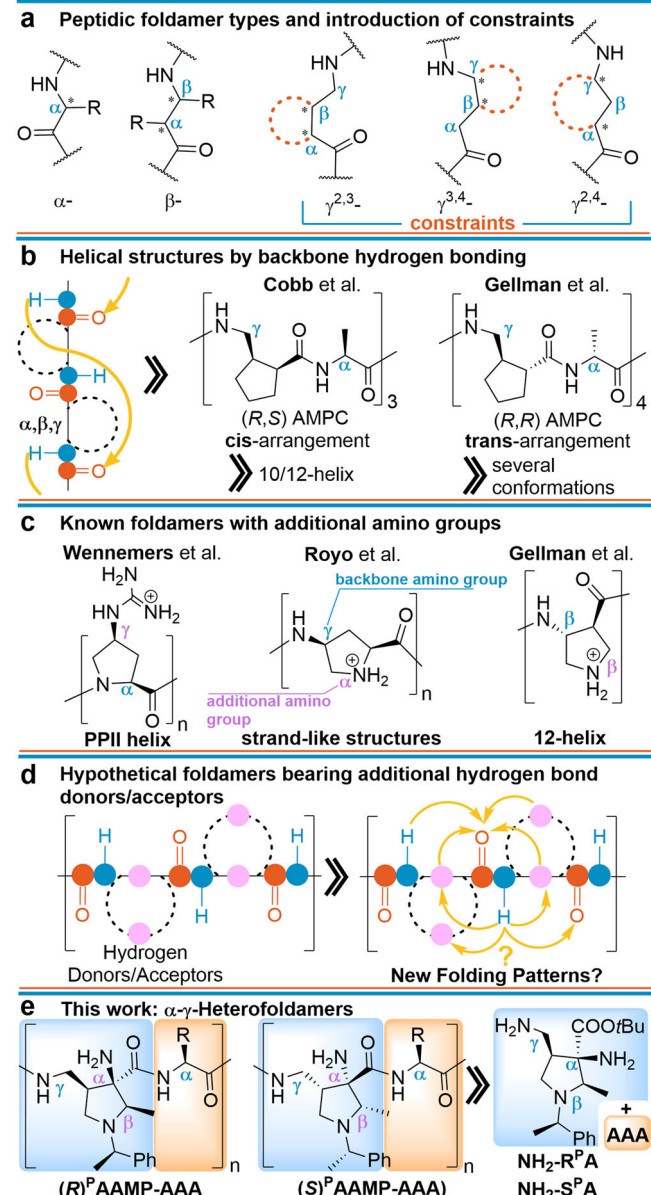

**Fig. 1 | Foldamers employing aliphatic amino acids. a** Foldamer types based on natural α-, non-natural β-, and γ-amino acid structural motifs including possible constraints in γ-amino acid-based foldamers. **b** Backbone hydrogen bonding and configuration influence. **c** Previous use of additional functionality in foldamers. **d** Potential secondary hydrogen bonding network by multifunctional building blocks. **e** Proposed foldamers based on α,β,γ-triamino acids (R)ᴾAAMP or (S)ᴾAAMP and α-amino acids.

stable foldamers. This is illustrated by Cobb's[43] and Gellman's[44] 2-(aminomethyl)cyclopentanecarboxylate (AMPC)-Ala heterofoldamers. Whereas the *cis*-AMPC-(S)-Ala hexapeptide forms a stable 10/12 helix, the *trans*-AMPC-(R)-Ala octapeptide displays a much weaker tendency to form ordered structures (Fig. 1b).

Although the introduction of heteroatom substituents may lead to opportunities for the discovery of new folding patterns, previous findings did not lead to fundamental changes in their secondary structures. Oxygen substituents additionally stabilize various turn structures by short-range interaction in short α- and α,β-peptides[45,46], β-strands[47], or contribute to backbone helical structures by weak electrostatic interactions[48,49]. The introduction of amino groups in the cyclic constrained backbone had no influence on the folding patterns compared to their parent foldamers. Wennemers et al.[50] demonstrated that γ-guanidinyl-functionalized

α-oligoprolines still favored a PPII helix conformation and confirmed that the attached positive charges do not affect the conformation of the backbone (Fig. 1c). Alternative γ-homofoldamers of the same building block investigated by Royo and coworkers formed strand-like structures in which the α-amino function was not involved as a structure-forming element[51]. Gellman and colleagues used γ-amino-β-prolines as foldamer building blocks and observed the same propensity to form a 12-helix as its carbocyclic analog without interaction of the ring nitrogen atom with the backbone[52]. The related *cis*-4-aminopiperidine-3-carboxylic acid (*cis*-APiC) also forms a helical 12/10-folding as demonstrated by Choi et al.[53]. These examples teach that a single additional heteroatom may not be sufficient for new hydrogen bond networks.

Nature teaches that natural proteins possess both rigid and dynamic regions in an aqueous environment[54,55], enabling the most efficient substrate binding. This principle is also evident in intrinsically disordered proteins, vital components of the cellular machinery, characterized by high mobility forming a quasi-continuum of rapidly interconverting conformations. Interestingly, they are characterized by a large share of charged and hydrophilic amino acids capable of forming hydrogen bond networks[56]. Nevertheless, it has been observed that most non-natural foldamer types adopt relatively rigid structures in nonpolar solvents[35,43,44], while their tendency for folding in aqueous environments is rather rare[33,34,50–53].

Guided by these facts, we hypothesize that non-natural amino acids, which provide additional nitrogen donor and acceptor sites forming strong hydrogen bonds, both at the periphery and along the backbone of foldamers, in combination with natural amino acids may significantly affect the folding patterns and result in unique secondary structure motifs (Fig. 1d). The design principle of sterically and hydrogen bond network-constrained polyfunctional γ-amino acids may enable helical arrangements with wider perimeter than the currently accessible 14–15-helical arrangements. This should consequently result in more dynamic behavior without losing folding propensity in aqueous media.

We report the synthesis of a family of foldamers consisting of (2R,3R,4R)-1-((R)-1-**p**henylethyl)-3-**a**mino-4-(**a**minomethyl)-2-**m**ethyl-**p**yrrolidine-3-carboxylates ((R)ᴾAAMP or their all-(S)-enantiomers (S)ᴾAAMP)[57,58] and α-amino acids (Fig. 1e). A simple removal of the peripheral phenylethyl group leads to water-soluble foldamers. Their secondary structure has been elucidated by a combination of X-ray crystallography, NMR, and both vibrational and electronic circular dichroism spectroscopies. The experimental results are further supported by molecular dynamics simulations and quantum chemical calculations.

## Results and Discussion

Since foldamer formation propensity and stability of the (ᴾAAMP-α-amino acid)ₙ oligomers may be influenced by their length[2] and the absolute configuration of AAMP relative to that of the applied α-amino acid (Fig. 1e), a modular solution-phase peptide synthesis approach from (R)ᴾAAMP-*t*Bu **NH₂-RᴾA** with either glycine or L-alanine, and from (S)ᴾAAMP-*t*Bu **NH₂-SᴾA** with L-alanine, respectively was pursued (Supplementary Figs. S1–S3). Three series of protected/unprotected (R = PhEt/R = H) hexamers **Pg-6GRᴾA/Pg-6GRᴴA** (i.e. **Fmoc-6**(hexamer)-[**G**ly-(R)-(N-ᴾAAMP)]₃), **Pg-6ARᴾA/Pg-6ARᴴA**, and **Pg-6ASᴾA/Pg-6ASᴴA**, and octamers **Pg-8GRᴾA/Pg-8GRᴴA, Pg-8ARᴾA**, and **Pg-8ASᴾA/Pg-8ASᴴA** were prepared (Fig. 2) and investigated for their secondary structures. Of those, octamer **Fmoc-8ARᴾA** irreversibly forms insoluble aggregates on evaporation, which disabled further investigations. This contrasting behavior highlights the importance of the relative configuration of units and the length of the oligomer. Furthermore, in order to clarify the impact of the steric constraints imposed by pyrrolidine rings on polyfunctional amino acids, truncated hexamer **Ac-6GSDab** was synthesized via solid-phase peptide synthesis method (Supplementary Information p. S76).

**Solid-state structure and NMR spectral investigation in solution**
Dimer **Fmoc-2GRᴾA** and tetramer **NH₂-4GRᴾA** were crystallized from ethyl acetate and dichloromethane in wet ether, respectively, and their

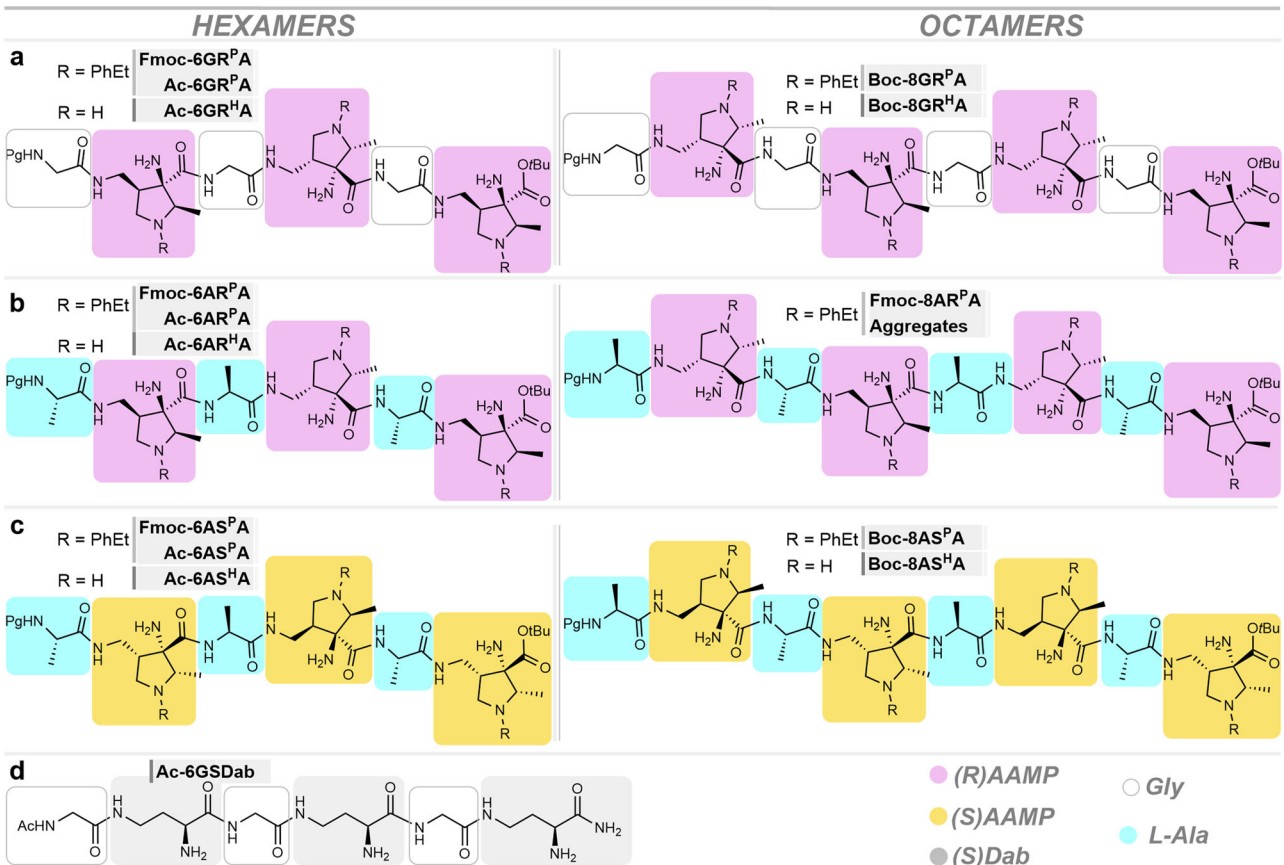

**Fig. 2 | Structures of α/γ-peptide hexamers 6 and octamers 8 containing AAMP and glycine or L-alanine units.** Foldamer series: (**a**) Pg[Gly-(*R*)AAMP]$_n$-O*t*Bu **Pg-6GRA** and **Boc-8GRA**; (**b**) Pg[L-Ala-(*R*)AAMP]$_n$-O*t*Bu **Pg-6ARA** and **Fmoc-8ARA**; (**c**) Pg[L-Ala-(*S*)AAMP]$_n$-O*t*Bu **Pg-6ASA** and **Boc-8ASA**; (**d**) Ac[Gly-(*S*)Dab]$_n$-NH$_2$ **Ac-6GSDab**; *n* = 3 or 4.

structure was determined by X-ray crystallography (Fig. 3). Intramolecular hydrogen bonds were not detected in the crystal structure of **Fmoc-2GR$^P$A** and the torsion angles ϕ, θ, ζ, and ψ of the (*R*)$^P$AAMP unit amounted to +83°, +60°, −94°, and −24°, respectively. In contrast, even though the crystal of **NH$_2$-4GR$^P$A** proved to be a hydrate (for the complete structure, see Supplementary Fig. S46), which may destroy potential intramolecular hydrogen bonds by forming its own hydrogen bonds to the carbonyl groups of both (*R*)$^P$AAMP units, the structure exhibited four intramolecular H-bond interactions. Importantly the structure revealed a hydrogen bond network between glycine amide N5 and the free NH$_2$ group (N4) at the pyrrolidine ring in a five-membered NH(*i*)···NH$_2$(*i*−1) hydrogen bond (2.66 Å, 111°), which is transmitted into an additional NH$_2$(*i*−1)···NR$_3$(*i*−1) hydrogen bond (2.85 Å, 104°) to the pyrrolidine nitrogen atom N3 (Fig. 3). This locks the geometry of the backbone in a bridged triazaspiranoid-like (i.e., ethano-bridged imidazoimidazole-like) arrangement, which changed the ϕ, θ, and ζ torsion angles moderately to +76°, +56°, and −91°, respectively, but ψ dramatically to +147°. Two more intramolecular hydrogen bonds involve the N-terminal glycine amino group N1 and the primary amino group N8 at the C-terminal pyrrolidine ring, both to the carbonyl oxygen atom O3 of the internal glycine unit, forcing the backbone to a turn-like arrangement. The latter two hydrogen bonds are unique to the tetramer, the higher oligomers adopt different hydrogen bond patterns (*vide infra*). The here found spiro/bridged hydrogen bond network is unknown. So far, five-membered hydrogen bonds have been recognized as conformationally constraining in planar arrangements between amide N-H and adjacent pyridyl nitrogen atoms in aromatic sp²-oligoamide foldamers[6,59], in so-called hydrazino turns between amide N-H functions and adjacent hydrazine nitrogen atoms[60–63], as well as in "fused-ring" hydrogen bonds between amide N-H functions and adjacent Lewis-basic oxygen atoms in the

backbone[64,65]. Larger six-membered hydrogen bonds are more common, e.g. in N-amino peptides leading to preferred β-sheet formation[66,67]. Despite all attempts using multiple solvents and crystallization conditions, crystals suitable for X-ray crystallographic structure determination of the hexamers and octamers were not successful since only microcrystalline solids were obtained.

The double five-membered hydrogen bond is also evident in the ¹H NMR spectra of tetramer **NH$_2$-4GR$^P$A** in CDCl$_3$ as well as in the N-terminal protected tetramer **Fmoc-4GR$^P$A**, where the internal glycine amide N5-H resonance is significantly downfield-shifted to 8.0 ppm, implying an internally H-bonded conformation in solution. In contrast, all other N-H protons are located upfield between 7.28 and 6.38 ppm. The bridging hydrogen bond causes deshielding by the pyrrolidine nitrogen atom, which is manifested by a slight downfield shift of the neighboring CH protons (3.18 ppm for CH-7 and 2.81/2.07 ppm for CH$_2$-5) compared to the non-H-bonded pyrrolidine-CH protons (3.03 ppm for CH-24 and 2.73/2.16 ppm for CH$_2$-22). A similar single downfield shift was even observed for a Boc-Gly-$^P$AAMP-Gly-O*t*Bu trimer **Fmoc-3GR$^P$AG**, the smallest oligomer capable of forming a three-dimensional bridged triazaspiranoid-like hydrogen bond network (Supplementary Information p. S64). The same patterns with respect to the chemical shifts of the C-H resonances next to the pyrrolidine nitrogen atoms were also found for all *N*-phenylethyl-protected hexamers **Fmoc-6$^P$** and octamers **Boc-8$^P$** for all pyrrolidine rings that can form the hydrogen bond network, but not for the C-terminal pyrrolidine rings, which are not able to form the hydrogen bond network. This was also not found for **Fmoc-6AR$^P$A**, which has a different solution structure (*vide infra*). However, this evidence remains indirect, since the NH$_2$(*i*−1)···NR$_3$(*i*−1) part of the spiro/bridged hydrogen bond network cannot be directly observed. In hexamers **Fmoc-6$^P$** and octamers **Boc-8$^P$**,

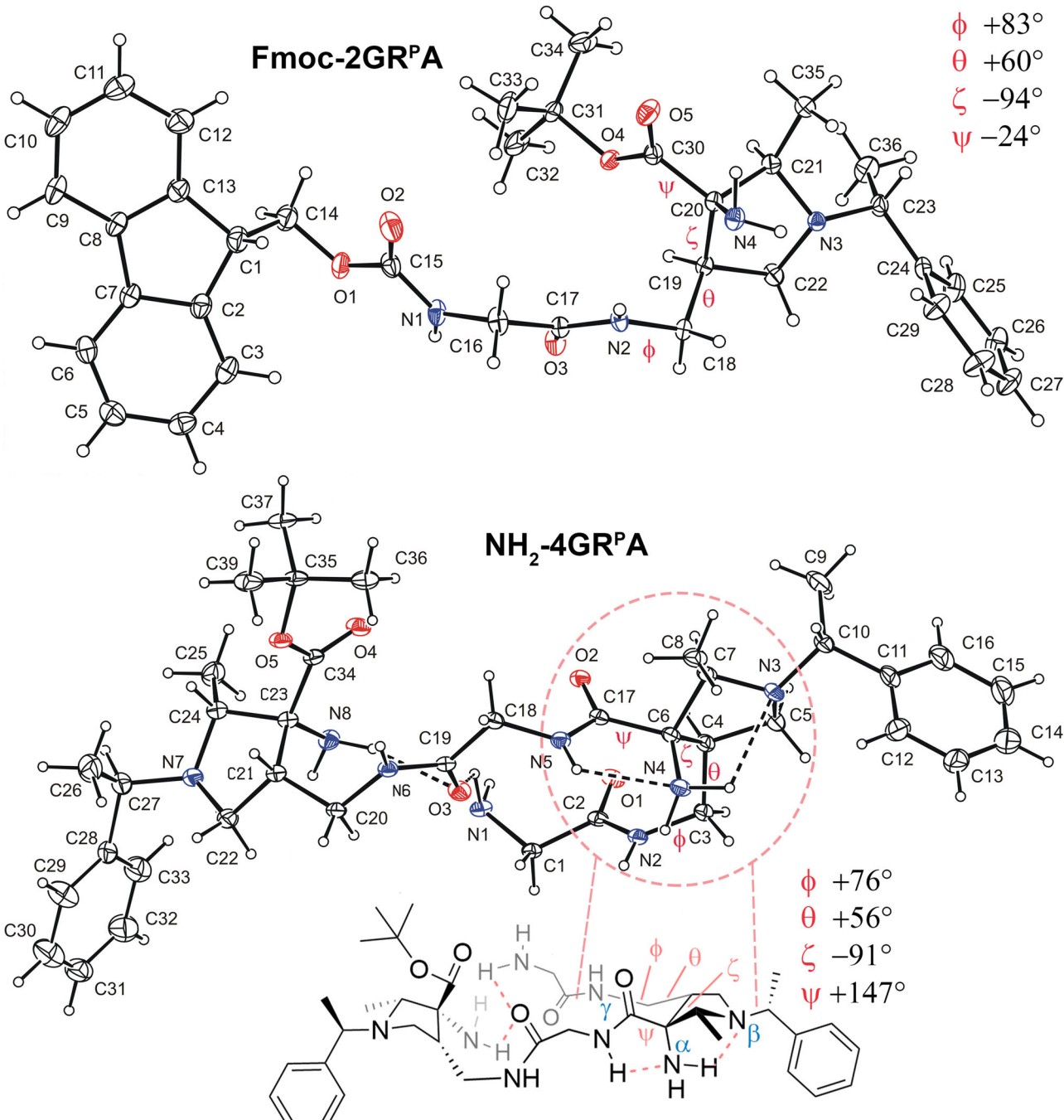

**Fig. 3 | X-ray crystal structures of α/γ-dimer Fmoc-2GR^PA and tetramer NH₂-4GR^PA.** The phenylethyl unit C27-C33 in **NH₂-4GR^PA** is disordered and only one orientation is depicted for clarity. The intermolecular interactions with water are omitted for clarity (for a complete structure see Supplementary Fig. S46). The displacement ellipsoids are drawn on 30% probability level for **Fmoc-2GR^PA** and on 20% probability level for **NH₂-4GR^PA**. Parameters of selected intramolecular hydrogen bonds in **NH₂-4GR^PA**: bond length X-H···Y (Å), angle at H(°): N1-H11···O3 2.900 (5), 123; N4-H41···N3 2.849(5), 104°; N5-H5···N4 2.660 (5), 111°; N8-H82···O3 3.260 (5), 164.

two downfield-shifted amide N-H resonances for **Fmoc-6^P** and three for octamers **Boc-8^P** were found at δ ~ 7.8–8.3 ppm, implying the presence of two or three double five-membered H-bonding NH(i)···NH₂(i−1)···NR₃(i−1) interactions (Supplementary Fig. S4). The other amide protons were, in contrast, found at δ ~ 7 − 8 ppm, representing the region associated with weakly H-bonded amides. The N-terminal carbamate N-H resonance is located upfield at 5.7–6.5 ppm, suggesting only weak participation in hydrogen bonding and consequently dynamic conformational behavior. Interestingly, all free NH₂ resonances occur at 1.89 ppm, suggesting a rather dynamic proton exchange.

The foldamers **6** and **8** can be diversified by simple hydrogenolytic removal of the phenylethyl groups, enabling comparison of both hydrophobic and hydrophilic character at the same backbone. Oligomers **Ac-6GR^HA, Ac-6AR^HA, Ac-6AS^HA, Boc-8GR^HA**, and **Boc-8AS^HA** with N-H pyrrolidine rings are indeed much more polar, basic, insoluble in CDCl₃, but reasonably water-soluble, allowing their structural investigation by NMR spectroscopy in H₂O/D₂O 9:1 solution with the pH adjusted to approximately 4.5 by addition of CD₃COOD. All amide N-H protons of **Ac-6^H** and **Boc-8^H** show chemical shifts δ > 8 ppm (Supplementary Fig. S6). Oligomers **Ac-6GR^HA** and **Boc-8GR^HA** display two out of six or three out of eight

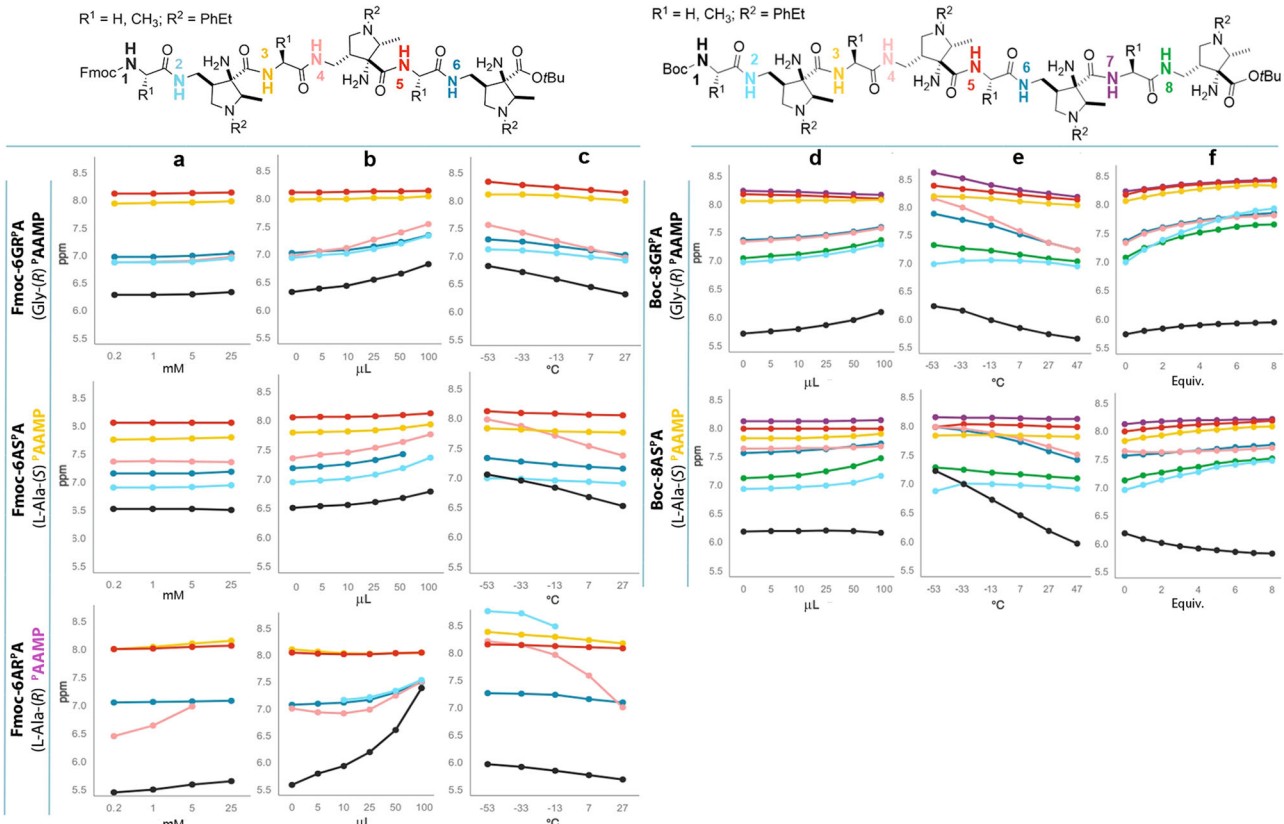

**Fig. 4 | $^1$H NMR spectroscopic investigation of hexamers Fmoc-6$^P$ and octamers Boc-8GR$^P$A and Boc-8AS$^P$A. a** Concentration dependence over the 0.2 to 25 mM range in CDCl$_3$. **b** DMSO titration by addition of 0, 5, 10, 25, 50, and 100 μL of DMSO to a 10 mM CDCl$_3$ solution. **c** Temperature dependence of the amide NH chemical shift in the range of –53 to 27 °C in 10 mM CDCl$_3$ solution. **d** DMSO titration by addition of 0, 5, 10, 25, 50, and 100 μL of DMSO to a 10 mM CDCl$_3$ solution. **e** Temperature dependence of the amide NH chemical shift in the range of –53 to 27 °C in 10 mM CDCl$_3$ solution. **f** CD$_3$COOD titration by addition of 1–8 equivalents to a 10 mM CDCl$_3$ solution.

amide N-H resonances at 8.7 ppm, respectively, which are those with 1,5-relation to the primary amine units, whereas the other N-H protons range from 8.1–8.3 ppm. This is similar to the corresponding resonances of protected oligomers **Fmoc-6GR$^P$A** and **Boc-8GR$^P$A** in CDCl$_3$, surprisingly suggesting that the primary amine functions are not protonated in the presence of CD$_3$CO$_2$D and the spiro/bridged NH(*i*)···NH$_2$(*i*–1)···NR$_3$(*i*–1) hydrogen bond network is preserved (*vide infra*). Hexamer **Ac-6AS$^H$A** and octamer **Boc-8AS$^H$A** exhibit more narrowly distributed N-H resonances but maintain the same order. The amide resonances of **Ac-6AR$^H$A** closely array within the range of 8.15–8.45 ppm.

Hexamer **Ac-6GSDab** lacking the conformationally constraining pyrrolidine rings was studied for comparison. The amide protons of Gly-L-Dab hexamer **Ac-6GSDab** are not visible in the $^1$H NMR spectrum, however, in aqueous acetic acid at pH 4.5 a similar chemical shift order of all amide protons as in **Ac-6GR$^H$A** was observed. Two of them were found at δ ∼ 8.9 ppm and the others ranged from 8.2–8.4 ppm (Supplementary Fig. S6). This indicates that the NH(*i*)···NH$_2$(*i*–1)···NR$_3$(*i*–1) hydrogen bond network may also exist in **Ac-6GSDab**, but this is not sufficient to form a defined secondary structure (*vide infra*).

**Secondary structure elucidation in solution**
**NMR titrations of hexamers 6 and octamers 8.** Having proved the existence of the hydrogen bond network formed by the additional amine functionalities in both water and nonpolar solvents, the identity of the secondary structure had to be determined. The concentration dependence of the amide N-H resonances over a 0.2 to 25 mM or 0.1 to 70 mM range in CDCl$_3$ revealed minimal chemical shift variation in hexamers **Fmoc-6GR$^P$A**, **Fmoc-6AS$^P$A** and octamer **Boc-8GR$^P$A**, respectively, indicating that *intramolecular* interactions are strongly predominant

(Fig. 4a and Supplementary Fig. S11). In contrast, the NMR experiments for **Fmoc-6AR$^P$A** show a significant downfield shift of the central N4-H proton (in pink) over the concentration range, accompanied by significant broadening and disappearance of the signal at the highest concentrations (25 mM). The N2-H proton (in light blue) was too broad to be detected. These results indicate that **Fmoc-6AR$^P$A** is more flexible and engages in concentration-dependent *intermolecular* interactions (*vide infra*, ROESY investigations).

The pyrrolidinecarboxamide protons N3-H (in yellow) and N5-H (in red) of all three hexamers **Fmoc-6$^P$** (Fig. 4b) experienced minimal changes on titration with the hydrogen bond interrupting co-solvent DMSO (5–100 μL)[23,39,43]. For hexamers **Fmoc-6GR$^P$A**, and **Fmoc-6AS$^P$A**, all other amide N-H protons experienced small downfield shifts of up to 0.5 ppm, indicating a slight weakening of intramolecular hydrogen bonding because of competitive interactions with DMSO; however, this change is too small to induce changes in the peptide secondary structure[39,43]. Even smaller effects were observed for all amide N-H protons in octamers **Boc-8GR$^P$A** and **Boc-8AS$^P$A** on titration with DMSO, suggesting increased conformational rigidity with increasing number of residues (Fig. 4d). In contrast, a large 2 ppm downfield shift of amide signal N1-H (in black), the appearance of the N2-H proton resonance only after the addition of larger than 10 μL DMSO, and the local minimum of the chemical shift of the N4-H proton (in pink) indicate flexible secondary arrangements at the N-terminal and central regions in hexamer **Fmoc-6AR$^P$A** (Fig. 4b, bottom).

Since the observed amide N-H shift values are weighted averages of the contributing hydrogen-bonded and non-hydrogen-bonded states, a larger downfield shift with decreasing temperature should be the result of longer time spent in the H-bond state. The temperature dependence of N-H resonances of hexamers **Fmoc-6GR$^P$A** and **Fmoc-6AS$^P$A** as well as

octamers **Boc-8GR$^P$A** and **Boc-8AS$^P$A** is minimal, especially for those involved in the five-membered hydrogen bond network, i.e. N3-H, N5-H and N7-H, respectively (Figs. 4c, e). The N-terminal N1-H (in black) and N4-H (in pink) in both hexamers and octamers, and N6-H (in blue) in octamers experienced a slightly larger 0.5 ppm downfield shift with decreasing temperature. Noticeably, the L-alanine-containing octamer **Boc-8AS$^P$A** appears to be more rigid than glycine-derived octamer **Boc-8GR$^P$A** since all the N-H proton resonances are less temperature-dependent, except for the N-terminal N1-H proton (in black) being largely flexible at room temperature. Conversely, hexamer **Fmoc-6AR$^P$A** displayed a significantly larger 1 ppm decrease of the N4-H proton shift (in pink) with increasing temperature; a similar trend was observed for the N2-H proton resonance (in light blue), which significantly broadened and disappeared at higher temperatures (7–27 °C) (Fig. 4c, bottom), suggesting more pronounced conformational flexibility along the backbone.

While DMSO minimally affected the solution structure of most hexamers and octamers, protonation of the amine functions might dramatically change it. However, the chemical shifts of amide protons surprisingly changed negligibly when the solutions of octamers **Boc-8GR$^P$A** or **Boc-8AS$^P$A** were titrated with acetic acid (Fig. 4f); the observed slight downfield shift indicated even stabilization of the structure. Surprisingly, the primary amine functions were apparently not protonated, potentially because of their five-membered hydrogen bonding with the nearby backbone amides, supporting a defined secondary structure. Protonation most likely occurs selectively at the more basic pyrrolidine nitrogen atoms; this is manifested by the significant 3–7 ppm downfield shift of the carbon atoms adjacent to the pyrrolidine nitrogen atoms in the $^{13}$C NMR spectra after addition of four equivalents of AcOH. Subsequent addition of another four equivalents resulted in only minor further 1–2 ppm downfield shift of the same carbon atoms (Supplementary Fig. S24). Additionally, the quaternary carbon atoms bearing the α-amino group displayed only 0.5–1 ppm downfield shifts after addition of the first four equivalents and no further shift after addition of another four equivalents. In contrast, the addition of only a single equivalent of methanolic HCl to **Boc-8GR$^P$A** or **Boc-8AS$^P$A** led to disappearance of all amide N-H resonances and appearance of unstructured broad bands, indicating highly dynamic systems (Supplementary Figs. S25, S26). Notably, trifluoroacetylation of the free α-amino groups resulted in similar $^1$H NMR spectra featuring unstructured broad bands, thus supporting the pivotal role of the hydrogen bond network involving the free α-amino groups.

In summary, the NMR titrations point to the presence of an intramolecular self-organized secondary structure in hexamers **Fmoc-6GR$^P$A**, **Fmoc-6AS$^P$A**, and octamers **Boc-8GR$^P$A**, **Boc-8AS$^P$A**, which becomes more stable with increasing length. The NH($i$)···NH$_2$($i$−1)···NR$_3$($i$−1) double five-membered hydrogen bonds formed by the additional amine substituents in the AAMP units serve as structure-forming elements and actively contribute to secondary structure generation, both in nonpolar and aqueous solution.

**ROESY investigations.** For all N-phenylethyl-protected hexamers and octamers, characteristic strong NOE contacts between the *trans*-oriented N-H proton of the α-amino acid ($i$) and the methyl group at the ($i$−1) pyrrolidine ring were detected, which result from the fixation of the amide proton via the five-membered NH($i$)···NH$_2$($i$−1) hydrogen bond in proximity to the methyl group (Fig. 5a–d, cf. Fig. 3). The involvement of all amino acid units in hexamers **Fmoc-6GR$^P$A**, **Fmoc-6AS$^P$A**, and octamers **Boc-8GR$^P$A**, **Boc-8AS$^P$A** in medium-to-strong ($i$ + 1), ($i$ + 2) and even more distant NOE contacts indicate the presence of highly populated conformations. For hexamers **Fmoc-6GR$^P$A**, **Fmoc-6AS$^P$A**, and octamers **Boc-8GR$^P$A**, **Boc-8AS$^P$A**, medium to strong NOE contacts between consecutive amide protons NH($i$) and NH($i$+1) exist, which were also observed by Gellman for ($R,R$)-AMPC-D-alanine oligomers (cf. Fig. 1b)[44]. For hexamers **Fmoc-6GR$^P$A** and **Fmoc-6AS$^P$A** strong ($i$ + 2) and even ($i$ + 4) NOE interactions between ($R$)$^P$AAMP(2) and ($R$)$^P$AAMP(4) as well as ($R$)$^P$AAMP(6), respectively, were observed (Fig. 5a, b), indicating a close spatial relationship of all three pyrrolidine

units. Although contacts between the pyrrolidine positions vary, they suggest similar structural motifs for both **Fmoc-6GR$^P$A** and **Fmoc-6AS$^P$A**. The close spatial relationship of pyrrolidine rings is further supported by the contact of the methyl group of the phenylethyl unit of N-terminal ($S$)$^P$AAMP to the CH proton of the phenylethyl unit of central ($S$)$^P$AAMP. Larger than ($i$ + 1) contacts were not observable for **Boc-8GR$^P$A** and **Boc-8AS$^P$A** because of overlapping signals of the $^P$AAMP residues (Fig. 5d).

A distinct pattern was found for the L-Ala-($R$)$^P$AAMP hexamer **Fmoc-6AR$^P$A**. Notably, there were no observed NOE contacts between consecutive amide protons NH($i$) and NH($i$ + 1) (Fig. 5c). Unexpectedly, strong NOE contacts between the N-terminal Fmoc protecting group and the C-terminal ($R$)$^P$AAMP(6) *tert*-butyl ester group were detected. This, together with backbone NOE contacts between the methyl groups of L-Ala units with the NH($i$ + 2) protons, may be traced either to an intramolecular turn-like structure or to intermolecular interactions. Since the NMR titrations of **Fmoc-6AR$^P$A** are highly concentration-dependent (cf. Figure 4), intermolecular association rather than a stable intramolecular, e.g. turn-like secondary structure, seems to be more likely for this oligopeptide.

The N-H pyrrolidine hexamers **Ac-6GR$^H$A**, **Ac-6AS$^H$A**, and **Ac-6AR$^H$A** were investigated in 9:1 H$_2$O/D$_2$O solution acidified by the addition of CD$_3$COOD to pH 4.5. Overlapping signals hampered the identification of the full NOE pattern, but several long-range interactions were unambiguously identified. Both **Ac-6GR$^H$A** and **Ac-6AS$^H$A** exhibit ($i$ + 3) contacts of the N-terminal α-amino acid units with the NH proton of the pyrrolidine aminomethyl group (Figs. 5e, f). An ($i$ + 4) contact of Gly(1) and Gly(5) is notable in **Ac-6GR$^H$A** (Fig. 5e), whereas ($i$ + 1) and ($i$ + 2) contacts prevail in **Ac-6AS$^H$A** (Fig. 5f). The observed long-range NOE contacts over the whole peptide backbone for hexamers **Ac-6GR$^H$A** and **Ac-6AS$^H$A** indicate ordered secondary structures; however, their characteristics differ from those of N-phenylethyl-protected **Fmoc-6GR$^P$A** and **Fmoc-6AS$^P$A** since NH($i$)···CH$_3$($i$−1) and NH($i$)···NH($i$ + 1) contacts are weak or absent. Thus, it is evident that the change in the solvent polarity and the peptide basicity results in a more variable population of secondary structures. For hexamer **Ac-6AR$^H$A**, ($i$ + 1), ($i$ + 2), and ($i$ + 3) contacts are limited to the central and C-terminal parts of the oligopeptide indicating large flexibility of the N-terminal amino acid units (Fig. 5g). The results of the ROESY investigation demonstrate that hydrophobic **Fmoc-6GR$^P$A**, **Fmoc-6AS$^P$A**, **Boc-8GR$^P$A**, and **Boc-8AS$^P$A** form relatively stable intramolecular secondary structures in nonpolar solvents. Hydrophilic oligomers **Ac-6GR$^H$A** and **Ac-6AS$^H$A** sharing the same backbone point to similar secondary structures in aqueous acidic solution, which seem to be, however, more dynamic. The ROESY results are a valuable basis for molecular dynamics simulations (*vide infra*).

**VCD and ECD study.** For more comprehensive elucidation of oligopeptide secondary structures, vibrational and electronic circular dichroism were used. ECD spectroscopy is commonly used for the structural determination of proteins, peptides, and their analogs[25,68,69], but its application for the foldamers bearing *N*-phenylethyl protection groups, which are only soluble in organic solvents such as CHCl$_3$ with a solvent-cut-off at ~230 nm is rather difficult. In contrast, VCD enables measuring all foldamer types[70–72], hence enabling structural and conformational information for peptides containing *N*-phenylethyl groups and comparing them to free N-H peptides.

Structural information is predominantly obtained by analysis of IR/VCD bands in the amide I and II spectral region (Fig. 6a–d). The IR amide I bands of the oligomers are observed in a spectral range of ~1650–1665 cm$^{-1}$. The amide II bands of phenylethyl-protected are found between ~1510–1520 cm$^{-1}$, whereas those of N-H pyrrolidines appear at a broader spectral range of ~1520–1555 cm$^{-1}$ (Fig. 6a–d upper part). The differences are likely caused by the use of CHCl$_3$ and MeOH as solvents, respectively.

The VCD spectra of phenylethyl-protected oligomers **Ac-6GR$^P$A**, **Boc-8GR$^P$A** as well as those of N-H oligomers **Ac-6GR$^H$A** and **Boc-8GR$^H$A**, display positive amide I couplets, implying a right-handed helical structure

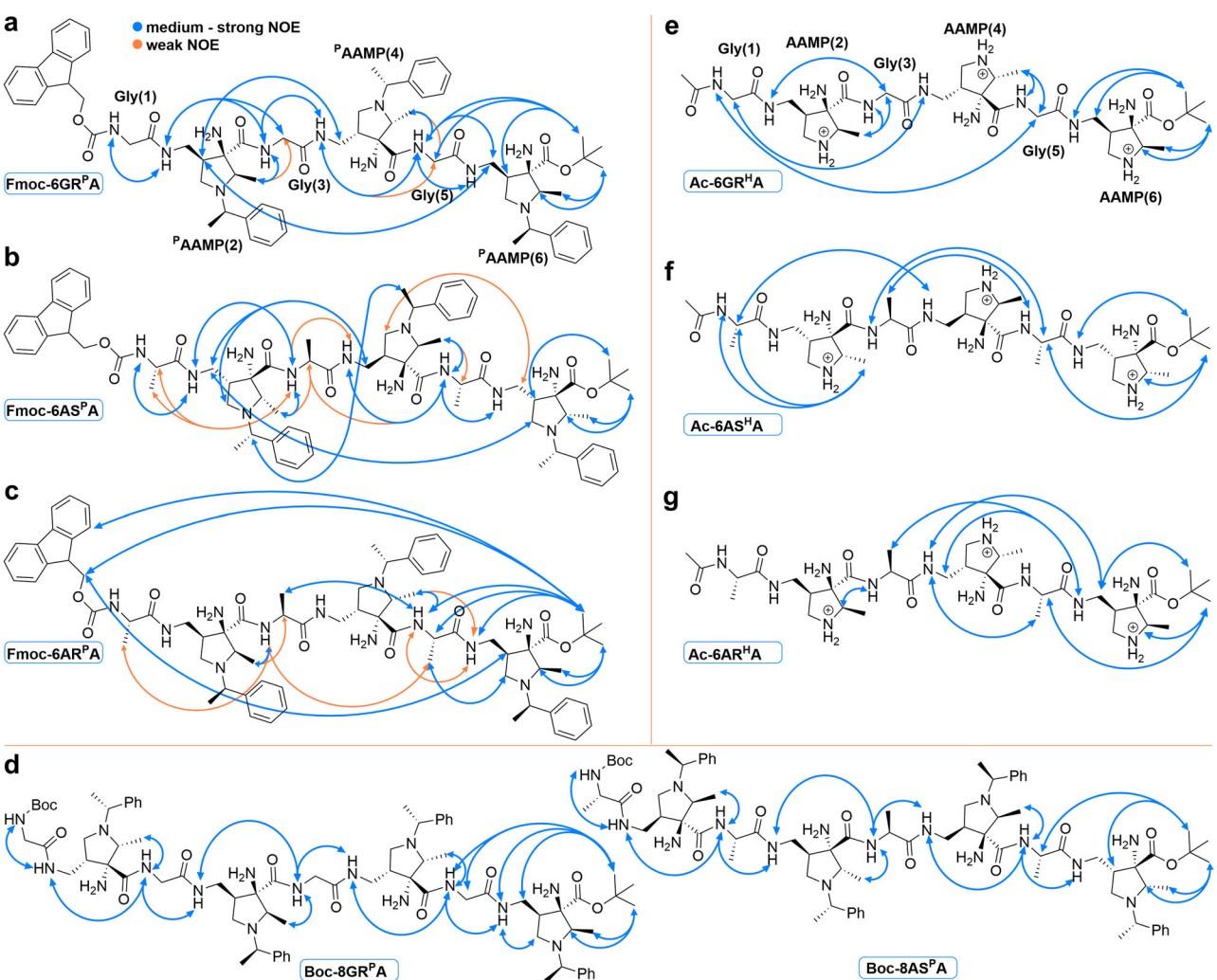

**Fig. 5 | Significant NOEs observed in ROESY experiments of hexamers and octamers. a** Fmoc-6GR[P]A; (**b**) Fmoc-6AS[P]A; (**c**) Fmoc-6AR[P]A hexamers; (**d**) Boc-8GR[P]A and Boc-8AS[P]A octamers in CDCl₃; (**e**) Ac-6GR[H]A; **f** Ac-6AS[H]A; and (**g**)

Ac-6AR[H]A hexamers in 9:1 H₂O/D₂O solution acidified by CD₃COOH to pH 4.5 (See Supplementary Figs. S29–36 for assignments of NOE cross-peaks).

(Fig. 6a–d lower part)[68,73,74]. In line with this result, oligomers Ac-6GS[P]A, Ac-6AS[P]A, Boc-8AS[P]A, and Boc-8GS[H]A display VCD spectra with negative amide I couplets, indicating a left-handed helical structure (Fig. 6a, blue and red, 6b-d blue)[68,73,74]. The lower intensity and broadening of the VCD bands of the N-H oligomers is probably caused by interactions with the polar solvent and their larger flexibility[75]. Despite the IR spectra of diastereomeric hexamers Ac-6AR[P]A and Ac-6AR[H]A are similar to those of their (S)-diastereomers (Figs. 6a, b), their VCD spectra display different patterns (Figs. 6a, b green). They are characterized by a +,−,+ amide I spectral pattern, indicating that these diastereoisomers have significantly different conformational preferences compared to all other oligomers.

For phenylethyl-protected α,γ-oligopeptides Ac-6GR[P]A, Ac-6AS[P]A, and Boc-8GR[P]A, Boc-8AS[P]A the amide A IR spectral region at 3200–3400 cm⁻¹ provides additional information about interactions of amide N-H functional groups (Supplementary Fig. S39)[76,77]. Infrared spectra of the compounds Ac-6GR[P]A, Ac-6GS[P]A, Ac-6AS[P]A, and Boc-8GR[P]A, Boc-8AS[P]A are characterized by two equally strong bands at ~3320 and ~3370 cm⁻¹, which represent similarly hydrogen-bonded N-H groups, and a shoulder at ~3450 cm⁻¹, which can be assigned to the NH₂ functional group. In contrast, the bands of Ac-6AR[P]A are more structured and of variable intensities, suggesting variable N-H hydrogen bonding.

ECD was only applicable for the N-H oligomers Ac-6[H], Boc-8GR[H]A, and Boc-8AS[H]A (Fig. 6e–j)[24,69]. Hexamer Ac-6GR[H]A with the (R)AAMP

unit is characterized by an intense negative maximum at ~203 nm with a negative shoulder at ~217 nm in MeOH, indicating a defined secondary structure (Fig. 6e)[26,52,78–80]. The negative band shifts to shorter wavelengths and increases in intensity with increasing solvent polarity. Furthermore, the negative shoulder at ~217 nm in MeOH changes to a positive spectral band at ~228 nm in water. A very similar ECD spectrum is observed for octamer Boc-8GR[H]A with a negative maximum at ~199 nm and a negative shoulder at ~217 nm in MeOH (Fig. 6f). However, no significant spectral shift is found in TFE and H₂O. The inversion of the negative shoulder to a positive band detected in TFE solution might be caused by additional intermolecular hydrogen bonds formed in the presence of TFE[80,81]. Hexamer Ac-6AS[H]A and octamer Boc-8AS[H]A with the opposite absolute configuration at the AAMP pyrrolidine ring and L-Ala are characterized by bisignate ECD spectra with a positive maximum at 201 nm and a negative one below 190 nm, indicating a similar, mirror-like arrangement compared to Ac-6GR[H]A and Boc-8GR[H]A (Fig. 6g, h)[52,78–80]. The ECD spectra of Ac-6AR[H]A show a negative spectral band at ~201 nm with a negative shoulder at ~222 nm, which is more prominent in TFE, whereas in H₂O a spectral shift to 199 nm with a sign-flip to a positive band at 222 nm was detected (Fig. 6i). Similar spectral patterns were found in the ECD spectra of all hexamers Ac-6[H] in aqueous acetic acid solution (Figs. 6e, g, i) compared to pure water suggesting that weak acids do not significantly influence their solution structure. The importance of the AAMP unit as secondary structure-

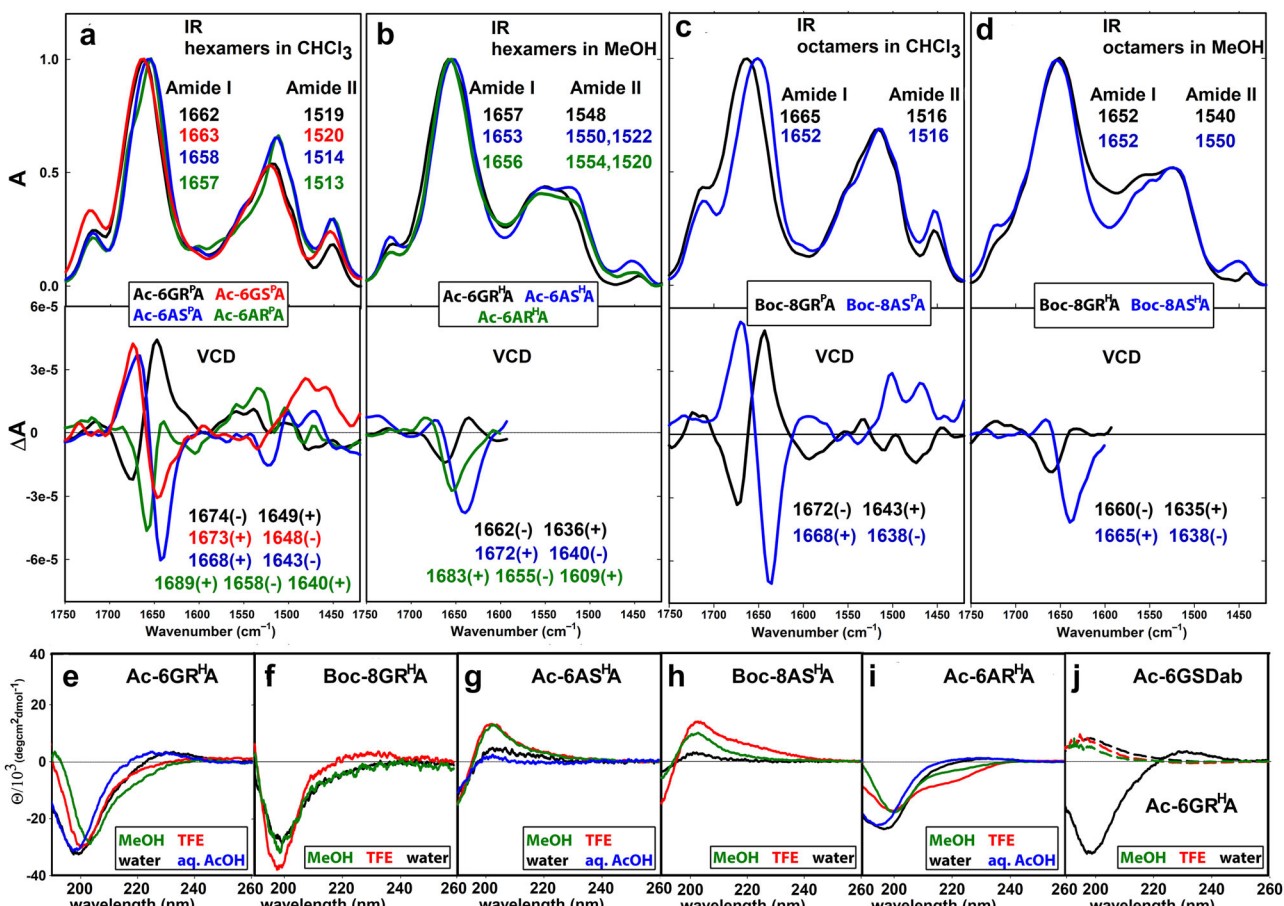

**Fig. 6 | IR/VCD and ECD spectra of the foldamers.** IR and VCD spectra of: (**a**) hexamers **Ac-6GR^PA** (black), **Ac-6AS^PA** (blue), **Ac-6AR^PA** (green), **Ac-6GS^PA** (red) in CHCl₃; (**b**) hexamers **Ac-6GR^HA** (black), **Ac-6AS^HA** (blue), **Ac-6AR^HA** (green) in MeOH; (**c**) octamers **Boc-8GR^PA** (black), **Boc-8AS^PA** (blue) in CHCl₃; (**d**) octamers **Boc-8GR^HA** (black), **Boc-8AS^HA** (blue) in MeOH (foldamer concentration 50 mM). (**e–i**) ECD spectra of foldamers **Ac-6^H** and **Boc-8GR^HA**, **Boc-8AS^HA** in MeOH (green), TFE (red), H₂O (black), and 240 mM aqueous acetic acid (blue) (foldamer concentration 1 mM); **j** Comparison of hexamers **Ac-6GR^HA** in water (black), and **Ac-6GSDab** in MeOH (dashed green), TFE (dashed red), and H₂O (dashed black).

forming element in the foldamers is evident by comparison with the ECD spectra of **Ac-6GSDab**; a single positive band at ~195 nm of rather low intensity (Fig. 6j) suggests an unordered structure of this oligomer.

The ECD results demonstrate that the spectral features of the peptide backbone are determined by the chirality of the AAMP units since the compounds containing the (*R*)AAMP units are characterized by a negative contribution at ~200 nm while those with (*S*)AAMP units exhibit bands of opposite sign. The lower intensity of the signal at ~200 nm for **Ac-6AR^HA** compared to **Ac-6GR^HA** can be likely traced to the spectral contribution of L-Ala.

Concerning the evolution of folding with increasing peptide length, VCD and ECD spectroscopy consistently showed that the dimer does not occupy a self-organized secondary structure, the tetramer shows developing spectral features indicating some degree of conformational organization, whereas the longer oligomers - hexamers and octamers clearly show the folding features discussed above (Supplementary Figs. S37, 38).

Taken together, VCD and ECD spectroscopy revealed that the secondary structures of all foldamers are dominated by the absolute configuration of the AAMP units, which possibly also induce the formation of more defined structures in solution. VCD spectroscopy allowed direct comparison of the structural arrangement of *N*-protected and free N-H foldamers in non-polar or polar solvents. The results indicate a left-handed helix as basic structural motif for the peptides containing (*S*)AAMP units and a right-handed helix for the foldamers containing Gly-(*R*)AAMP units; the hexamers **Ac-6AR^PA** and **Ac-6AR^HA** are in contrast structurally different. The ECD and/or VCD spectra demonstrate that the elongation of the

foldamer does not lead to a significant change in the peptide secondary structure. The spectral variance in different solvents can be traced to variable solvent-solute interactions, causing changes in the compactness of foldamer molecular structure and flexibility, but the handedness remains conserved.

## Computational Investigation of Foldamers Fmoc-6GR^PA, Ac-6GR^HA and Boc-8GR^PA

To further characterize the secondary structures of the hexamers and octamer containing Gly-(*R*)^PAAMP unit (**Fmoc-6GR^PA, Ac-6GR^HA, Boc-8GR^PA**), we resorted to computational methods. Molecular dynamics simulations of the foldamers provide reliable information about molecular contacts and structural diversity. In addition, quantum-mechanical calculations allow the computation of NMR shifts of selected atoms in different chemical environments. The combination of these two techniques, when interpreted constrained with the above experimental observations, provides molecular-level information of the accessible conformations and preferential secondary structure (see Supplementary information p. S44–50 for methodological details). Each potential structural model of the experimentally studied foldamers is denominated with the prefix M and with information of the simulation details in parenthesis at its end, e.g., **MFmoc-6GR^PA**(RH) is a computational model of hexamer **Fmoc-6GR^PA** obtained from a (R)ight-(H)anded helical structure simulation (see Supplementary Data 1–5 for detailed 3D visualization of the representative structures). We focus on the NH(i)⋯NH2(i−1)⋯NR3(i−1) interactions as they are the prevalent features in experiments and in our simulation models.

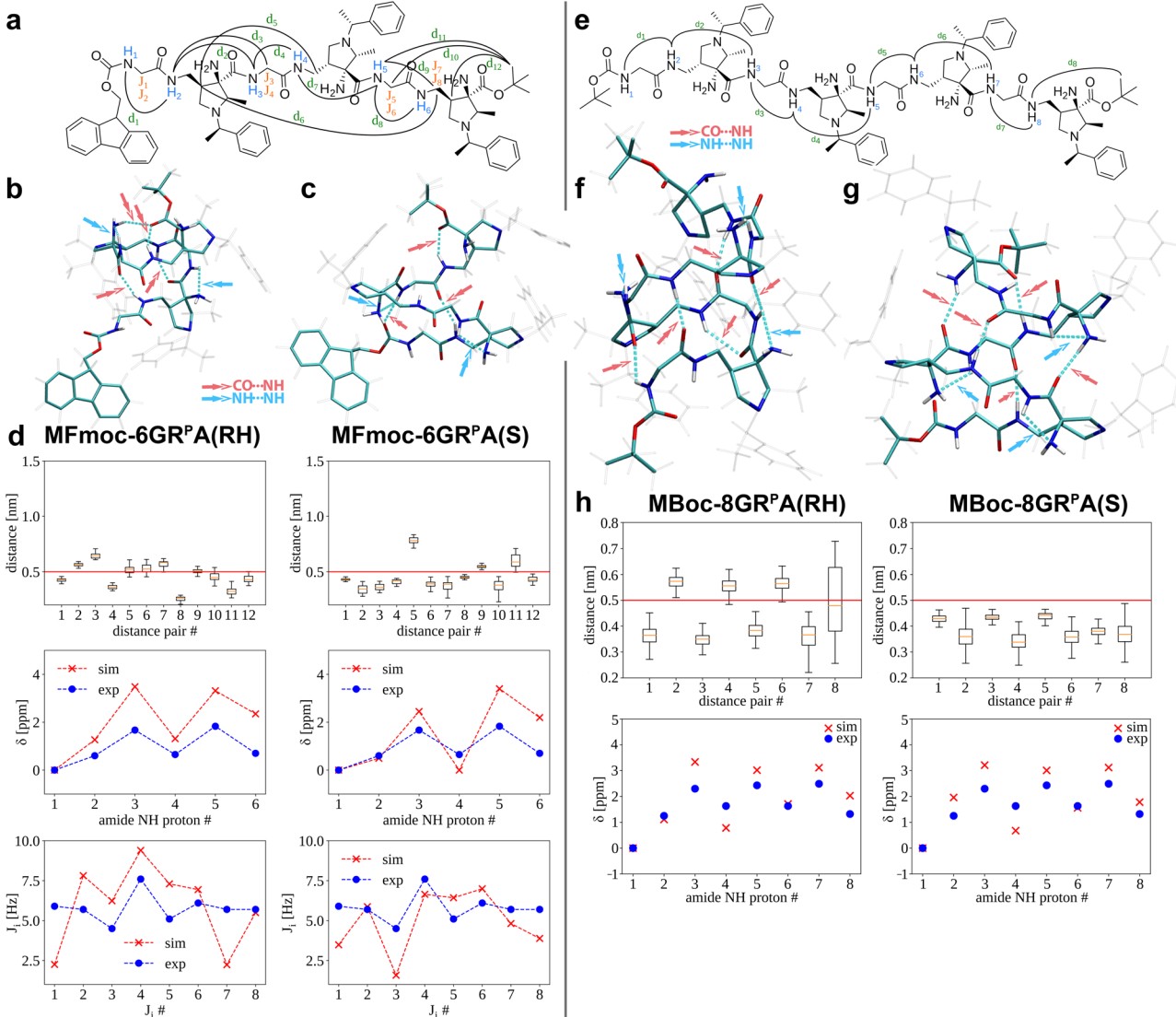

**Fig. 7 | Visualization of representative structures and comparative analysis of experimental and simulated data. a** Experimental long-range NOEs $d_i$ (cf. Figure 5a), observed amide proton chemical shifts $\delta_i$, and spin-spin couplings $J_i$ of **Fmoc-6GR^P A** used for comparing to simulation data. **b** Representative structures of right-handed helical structure **MFmoc-6GR^P A**(RH) and (**c**) Left-handed staircase-like structure **MFmoc-6GR^P A**(S) resulting from unrestrained MD simulation. **d** Comparison of simulated NOEs (distances), amide proton chemical shifts, and spin-spin coupling constants with comparison to experimental observables. **e** Experimental long-range NOEs $d_i$ (cf. Figure 5d) and amide proton chemical shifts $\delta_i$ of **Boc-8GR^P A** used for comparing simulation data. **f** Right-handed helical

structure **MBoc-8GR^P A**(RH) and (**g**) Left-handed staircase-like structure **MBoc-8GR^P A**(S). **h** Comparison of simulated to experimental NOEs and NH amide proton chemical shifts. In the structures hydrogen bonds for these representative conformers are shown in dashed cyan lines. Blue (NH-NH) and red (NH-CO) arrows indicate the nature of the hydrogen bond. The red line in NOEs represents the 0.5 nm observable distance threshold. The box plot demonstrates the locality through their quartiles, from the lower quartile (Q1) to the upper quartile (Q3) and features a median line in red. Whiskers stretch out from the box to the most distant data point, which is no more than 1.5 times the interquartile range away.

Starting with the hexamer **Fmoc-6GR^P A**, an unrestrained MD simulation of a single foldamer solvated in chloroform was performed. Three different representative structural models were studied: 1) The average structure of the unrestricted (U) simulation – **MFmoc-6GR^P A**(U) (Supplementary Fig. S42) obtained by selecting a structure every 100 ns during the unrestricted simulation; 2) A cluster of right-handed (RH) helical structures – **MFmoc-6GR^P A**(RH) (Fig. 7b) obtained by a short simulation of a potentially contributing right-handed helical structure; and 3) A cluster of left-handed staircase (S) structures – **MFmoc-6GR^P A**(S) (Fig. 7c) obtained by a short simulation of a left-handed staircase-like structure observed during the unrestrained simulation. These two model structures were selected since they formed a secondary structure and were observed multiple times along the simulation trajectory constituting a significant part

of the unrestricted MD simulation, i.e. 15% and 29% in **MFmoc-6GR^P A**(RH) and **MFmoc-6GR^P A**(S) model structures, respectively.

For each cluster, the experimentally observed NOE contacts d1-d12, amide proton chemical shifts δ1-δ6, and spin coupling constants J1-J8 were calculated (Fig. 7d). The **MFmoc-6GR^P A**(RH) cluster model, which represents a right-handed helical structure with a well-defined hydrogen bond network, results in a good agreement of calculated chemical shifts with the experimental data. The trend of spin-spin coupling constants is well reproduced, but larger deviations at the N- and C-terminal ends of the helical structure compared to the experimental results are found. Moreover, some NOE contacts, e.g., d3 and d7, are on the borderline of being observable (>0.5 nm). The **MFmoc-6GR^P A**(S) cluster model represents a left-handed motif held together by a well-defined hydrogen bond network, corresponding to an unusual shape referred to here as a staircase. In this

cluster the amide N4-H chemical shift is smaller than experimentally observed, being similar to the N1-H value because it has no hydrogen bonding. Several spin-spin couplings show larger deviations from the experimental values, especially J1, J7, and J8 at the C- and N-termini, but also J3. Moreover, the simulated d5 NOE distance is too large to be experimentally observed (>0.5 nm). Thus, the computational results for **Fmoc-6GR$^P$A** do not point to a single populated structure in CHCl$_3$. On the other hand, the computationally found right-handed helical structure **MFmoc-6GR$^P$A**(RH) provides reasonable agreement with experimental NMR, ECD, and VCD data and therefore it may be considered as the prevailing one.

Octamer **Boc-8GR$^P$A** solvated in chloroform was investigated similarly as **Fmoc-6GR$^P$A**; however, less experimental data were available for comparison (Fig. 7e). Therefore, a biased MD simulation was initially used to generate an extended model structure **MBoc-8GR$^P$A**(E) (Supplementary Fig. S43). For this model, the simulated values are too distant compared to the experimental data indicating the formation of a secondary structure. Based on this, an unrestrained MD simulation revealed two structural motifs similar to those observed for **Fmoc-6GR$^P$A**, i.e., a right-handed 18/20-helical structure **MBoc-8GR$^P$A**(RH) (Fig. 7f) and a left-handed staircase-like structure **MBoc-8GR$^P$A**(S) (Fig. 7g). These structures account for 51% (**MBoc-8GR$^P$A**(RH)) and 4% (**MBoc-8GR$^P$A**(S)) of the simulation time. The simulated helical structure **MBoc-8GR$^P$A**(RH) shows reasonable agreement of the NOE contacts, although d2, d4, and d6 are on the borderline of being observable in the simulated structure, but still possible considering the statistical variability of the data (Fig. 7h). **MBoc-8GR$^P$A**(S) reproduces all observed experimental NOE data. The simulated and experimental amide chemical shifts fit well for both, **MBoc-8GR$^P$A**(RH) and **MBoc-8GR$^P$A**(S); therefore, both simulated secondary structure motifs are feasible within the given NMR experimental restraints. Quantum mechanical calculation of the relative energies ($\Delta E = E_{RH}-E_S$) of **MBoc-8GR$^P$A**(RH) and **MBoc-8GR$^P$A**(S) revealed a small difference of $\Delta E = -4.6$ kJ/mol in favor of the right-handed **MBoc-8GR$^P$A**(RH); however, this value has to be taken with caution based on the accuracy of the simulation and its disregard for the entropic contribution. Since chiroptical investigations (*vide supra*) point to a right-handed helix as the basic structural motif for the foldamers containing L-amino acid-(R)$^P$AAMP units, model **MBoc-8GR$^P$A**(RH) represents a viable structural candidate.

Finally, the fully protonated hexamer **Ac-6GR$^H$A** solvated in water was also evaluated using a non-biased MD simulation as for **Fmoc-6GR$^P$A** and compared to the experimental results (Supplementary Fig. S44). Filtering the simulation results by NOE distance restraints yields right-handed model structures compatible with the VCD expectation; however, the values of the calculated coupling constants considerably deviate from the experimentally determined. One potential reason may be the full protonation state of the MD simulation model, while the experimental investigation speaks rather for partial protonation (cf. Fig. 4f and Supplementary Fig. S27), which is not possible using our computational methodology. Nevertheless, our simulation and experimental data suggest that folding in **Ac-6GR$^H$A** occurs, but the overall structure is more flexible in aqueous solution and several conformations are likely populated.

## Conclusion

Peptide-like foldamers containing the polyfunctional α,β,γ-triamino acids AAMP (3-**a**mino-4-(**a**minomethyl)-2-**m**ethyl**p**yrrolidine) were synthesized and their secondary structure was determined. We demonstrate that both free α- and β-amino groups present in AAMP contribute to secondary structure formation in α,γ-oligopeptides and induce specific folding patterns. Crystallographic data of the Gly-(R)$^P$AAMP tetramer reveal a hydrogen bond network featuring double five-membered NH(i)···NH$_2$(i–1)···NR$_3$(i–1) interactions that prove crucial as secondary structure inducing elements. These key interactions are also supported by NMR studies. The data show a comparable intramolecular structural behavior of Gly-(R)$^P$AAMP and L-Ala-(S)$^P$AAMP hexamers and octamers

(**GR$^P$A** and **AS$^P$A** series), respectively. The relatively stable secondary structures of all oligomers containing AAMP units are supported by both VCD and ECD spectroscopies. VCD investigations point to a dominant right-handed helical arrangement of Gly-(R)AAMP α,γ-oligopeptides and a left-handed helix in the corresponding L-Ala-(S)AAMP and Gly-(S)AAMP α,γ-peptides. Importantly, VCD spectroscopy allowed the direct comparison of the structural arrangement of all peptides in polar and nonpolar environments. On the other hand, ECD spectroscopy provided evidence that the helical arrangement is preserved even in acidified aqueous solution using weak acids. NMR titration with acetic acid indicates selective protonation on the pyrrolidine nitrogen atoms, which does not lead to significant changes in the mean secondary structure. Molecular dynamics simulations for Gly-(R)AAMP α,γ-peptides reveal preferential occupation of two secondary structures with opposite handedness; of those the right-handed structure is in agreement with the results of the chiroptical investigations. For the free NH α,γ-peptide Gly-(R)AAMP **Ac-6GR$^H$A**, the NMR and VCD study points to a similar, but more dynamic structure, and the MD simulation for **Ac-6GR$^H$A** also suggests a more flexible overall structure in aqueous solution.

Our study supports the hypothesis that extra amino groups dominate the H-bonding pattern via NH(i)···NH$_2$(i–1)···NR$_3$(i–1) arrangements and play an equally important role as common peptide amide backbone hydrogen bonding and conformational constraints in steering the overall secondary structure. Since similar structures were found for peptides with both flexible glycine units and with helix-inducing alanine units, we anticipate that combination of AAMP units with arbitrary natural amino acids allows for obtaining 18/20 helical arrangements. Furthermore, the described spatially wider 18/20 helix might potentially serve as a helical host, accommodating molecules of complementary size and functionality. Thus, in a more general context, extra functional groups with hydrogen bonding capabilities may be considered as a step towards encoding and enhancing specific folding propensities in peptides and thus open new directions in engineering of bio-inspired materials.

## Methods
### General information for oligomers synthesis

Reactions not involving aqueous conditions were performed in flame-dried glassware under an argon atmosphere. Solvents and additives were dried prior to use according to standard procedures. TLC analyses were performed on POLYGRAM SIL G/UV254 plates. Chromatographic separations were carried out on silica gel 60 (Fluka, 230–400 mesh) either manually or on a CombiFlash® NextGen 300+ instrument. Full details of the chemical synthesis and purification are given in the Supplementary Information p. S4–7 and S58–98. For NMR spectra of the synthesized compounds, see Supplementary Data 6.

### X-Ray crystallography

Dimer **Fmoc-2GR$^P$A** was crystallized from EtOAc using slow evaporation at room temperature overnight. Tetramer **NH$_2$-4GR$^P$A** was crystallized from dichloromethane/wet ether (1:1, v/v) using slow evaporation at room temperature over 2 days.

Diffraction data sets of **Fmoc-2GR$^P$A** were collected on a Bruker D8 VENTURE Kappa Duo diffractometer with a PHOTON100 detector with micro-focus sealed tube CuKα (λ = 1.54178 Å) x-ray source IμS at 130 K. Diffraction data sets of **NH$_2$-4GR$^P$A** were collected on a Bruker D8 VENTURE Kappa Duo diffractometer with a PHOTONIII detector with micro-focus sealed tube CuKα (λ = 1.54178 Å) x-ray source IμS at 120 K. Details of data collection and refinement are given in the Supplementary Information p. S51–57.

### NMR spectroscopy

$^1$H and $^{13}$C NMR spectra were recorded on a Bruker Avance III™ 400, 500, or 600 spectrometers operating at 400, 500, or 600 MHz for $^1$H NMR and 100.1, 125.7, or 150.9 MHz for $^{13}$C NMR. Temperature-dependent spectra were recorded on a Bruker Avance II™ 500 MHz instrument. Full

assignment of $^1$H and $^{13}$C signals was achieved by a combination of 2D experiments ($^1$H,$^1$H-COSY; $^1$H-$^{13}$C HMBC; $^1$H-$^{13}$C HSQC). Spatial long-range contacts were determined by $^1$H,$^1$H-ROESY experiments. CDCl$_3$ was dried over 3 Å molecular sieves prior to the measurements. Measurements in H$_2$O/D$_2$O were performed using selective presaturation to suppress the H$_2$O signal. For detailed NMR study and 2D NMR spectra, see Supplementary information p. S8–36.

## CD spectroscopy

ECD experiments were carried out on a J-1500 spectropolarimeter (Jasco, Tsukuba, Japan). The spectra were collected from 180 to 280 nm at room temperature in 0.01 cm cylindrical quartz cells at 1 mM concentration in H$_2$O, TFE, or MeOH using the following setup: 5 nm/min speed, 16 s time constant, 1 nm spectral bandwidth, 2 scans, 0.5 nm steps. After baseline subtraction, the final data were expressed as molar ellipticities θ (deg·cm$^2$·dmol$^{-1}$) per residue.

VCD spectra were recorded on a commercial dual source VCD spectrometer (Chiral*IR-2X*™, BioTools, Inc., U.S.A.) working in a dual PEM mode using two ZnSe photoelastic modulators (36.996 and 37.02 kHz, Hinds Instruments, Inc., U.S.A.). The VCD data were collected for ~12 h (12 blocks of 6000 scans each at 8 cm$^{-1}$ resolution) at room temperature. A CaF$_2$ cell with 0.025 mm path length (SpeCac) was used for samples dissolved in MeOH and a NaCl cell with 0.1 mm path length (SpeCac) was used for samples dissolved in CHCl$_3$. For both methods, solvent scans were subtracted as background. The baseline was corrected using a linear function. Final IR spectra were normalized to amide I intensity maxima. The VCD spectra were smoothened with a second-order Savitzky-Golay filter using a 9 point window and normalized to amide I maxima in the corresponding IR spectra. Numerical data treatment was carried out using the Grams/AI software (Thermo Electron, Waltham, MA, USA).

For further details associated with the CD study, see Supplementary Information p. S37–43.

## Computational methods

For detailed information associated with the computational simulations, see Supplementary information p. S44–50.

## Data availability

Detailed 3D visualizations of the representative structures are available as Supplementary Data 1–5. The source datasets for molecular dynamics simulations and quantum chemical calculations have been deposited in the Zenodo repository and can be accessed via https://doi.org/10.5281/zenodo.10816862. All NMR spectra of synthesized compounds are provided in Supplementary Data 6. The X-ray crystallographic coordinates for structures reported in this Article have been deposited at the Cambridge Crystallographic Data Centre (CCDC), under deposition numbers CCDC 2142444 (**Fmoc-2GR$^P$A**) and 2142445 (**NH$_2$-4GR$^P$A**). These data can be obtained free of charge from The Cambridge Crystallographic Data Centre via www.ccdc.cam.ac.uk/data_request/cif. The CIF files are included as Supplementary Data 7, 8. All other data supporting the findings of this study are available in the article and Supplementary Information files and are additionally available from the corresponding authors on request.

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

## Acknowledgements

This work was generously funded by the Institute of Organic Chemistry and Biochemistry of the Czech Academy of Sciences (RVO: 61388963), the Gilead Sciences & IOCB Research Center and the European Regional Development Fund, OP RDE, Project: Chemical biology for drugging undruggable targets (ChemBioDrug) (No.CZ.02.1.01/0.0/0.0/16_019/0000729). I.C. thanks the Ministry of Education, Youth and Sports of the Czech Republic (MSM0021620857) for financial support. We thank Prof. Dr. Klaus Ditrich, BASF SE for a gift of 1-phenylethylamines.

## Author contributions

Conceptualization: D.J., U.J. Experimental Investigation: D.J., K.B., L.B., M.P., I.C. Computational Investigation: V.P., H.M.S. Supervision: U.J., H.M.S. Writing: D.J., V.P., K.B., L.B., M.P., I.C., H.M.S., U.J.

## Competing interests

The authors declare no competing interests.

## Additional information

**Peer review information** : *Communications Chemistry* thanks the anonymous reviewers for their contribution to the peer review of this work. A peer review file is available.

