## [Peer Review File · Communications Chemistry]

Reviewers' comments:

Reviewer #1 (Remarks to the Author):

This paper describes the study of an alpha,gamma-diamino-pyrrolidine monomer within a heteromeric 1:1-foldamer systems with alpha-amino acid residues. It is a thorough piece of work, where the major novelty is the structural consequence of a hydrogen bond between the primary amine of the alpha-amino-pyrrolidine and the nearby amide bond. The evidence of this comes from a combination of elucidation techniques such as XRD, NMR and CD. I think the authors should explain why there is not a crystal structure for anything above the tetramer though. Unfortunately the tetramer XRD itself cannot be extrapolated to the longer structures, but the key feature of the intramolecular NH₂--HNC(O) has been observed by NMR and predicted by MD. Nevertheless there are still two possible structures predicted for the octameric 8RPAG-Boc that fit the data. Whilst the evidence leans towards the right handed helix, the energy difference is only 4.6 kJ/mol. This is quite remarkable and further highlights what a shame it is that XRDs of the longer structures were not obtained (indeed could the existence of two energetically folds be the reason?). The observation that this secondary structure appears to remain even in weak acid is very interesting. The work done is exactly as you would expect for the determination of secondary structures. DMSO titrations, MD simulations based on nOEs, CD (it would be nice to see a side by side comparison of the CD of tetramer through to octamer).

In conclusion it is a nice study of quite an interesting monomer which makes an apparent foldamer with very interesting properties (particularly in terms of the stability in polar solvents and weak acid). Whilst the evidence for helix formation is very strong, the exact nature of it is inconclusive. The intramolecular bond is certainly a highlight (one wonders what the effect of dimethylating (or monomethylating) this amine would be? Would there be a different helicity?). Ultimately I am supportive of the submission being accepted with the minor edits below.

(1) I think the biggest problem is how long this report is for a communication. I would recommend really condensing the discussions about synthesis (Fmoc solubility etc), and just highlighting the most important DMSO titration graphs, CD, nOEs etc. There really is far too much detail in there that could go to the SI.

(2) Figure 2, though colourful is really difficult to assimilate. I'm not sure I would include the synthesis anyway (see above) and just show the structures that have been made/examined in a normal figure.

(3) In figure 3, I think the full chemical structure of 4RPAG-NH₂ should be presented to show all H-bonds including O3--N8

(4) Resolution of Figures 4,6 and 7 are poor.

Reviewer #2 (Remarks to the Author):

The authors designed, synthesized and characterized several hybrid α/γ peptidic oligomers. The design principle is that by adding steric and hydrogen bonding functional groups to the backbone or periphery of the cyclic γ peptide unit, and with an alternating α/γ sequence, may lead to new folding pattern and secondary structure. The authors were able to synthesize several different variation of the hybrid

foldamers with up to octamer, and crystalized dimer and tetramer. Then using a combination of experimental (crystallography, NMR, VCD/ECD) and computational (MD, DFT) methods, the authors characterized the structure of the oligomers and analyzed how several factors such as amino acids used (alanine or glycine), terminal groups, and chirality affects the secondary structure formation of the foldamer.

The reviewer believes this topic is of great interest to researchers in the field of foldamer design and organic synthesis and characterization and also should be of interest to general scientific communities. The design is novel and synthesis is sound. The authors also employed ample characterization methods to reveal and corroborate structural information obtained from different methods. The reviewer considers the quality of the work as very good.

However, the reviewer has the following concerns and hope the authors can address further:

1. Crystal structure of tetramer shows a $\text{NH}(i)\cdots\text{NH}_2(i-1)\cdots\text{NR}_3(i-1)$ hydrogen bonding network. The author concludes that this hydrogen bond chain is the key structural feature that induces the helical folding.
2. In the NMR spectral analysis (chemical shifts, titration, and Rosey), the focus seems to be just on the amide NH proton. What about the protons on the NH_2 group that is on the α carbon of the γ -amino acid unit, given the focus in the design on the γ -amino group?
3. The results from NMR/VCD etc. may support the $\text{NH}(i)\cdots\text{NH}_2(i-1)\cdots\text{NR}_3(i-1)$ hydrogen bonding network but it is not direct. Is it possible there is another structural feature that are also consistent with the NMR/VCD data? Does the MD generated structures contain this hydrogen bond chain?
4. The MD simulation structural figures in Figure 7 clearly shows hydrogen bonds marked as light blue dashed lines. These are mostly hydrogen bonds between amide or amino NH with carbonyl O in the peptide bond. However, there is no discussion about this types of hydrogen-bonds. Any particular reason?

Reviewer #3 (Remarks to the Author):

The manuscript entitled "Foldamers Controlled by Functional Triamino Acids: Design and Structural Investigation of α/γ -Hybrid Peptide Oligomers" submitted for publication in Communications Chemistry by Dr Jahn and colleagues, describes the synthesis and conformational studies of hybrid γ,α -peptide foldamers consisting of alternating 3-amino-4-(aminomethyl)-2-methylpyrrolidine-3-carboxylate (AAMP) and natural amino acids glycine or alanine. The study of the preferred secondary structures of these new foldamers is explored in solid state by X-ray analysis (of a dimer and a tetramer) and in solution by NMR spectroscopy (1D VT NMR, concentration and pH variation, DMSO titration, acid titration, 2D NMR, ROESY), chiroptical methods (VCD, ECD), and molecular dynamics simulations with and without NMR constraints. These foldamers incorporating α,β,γ -triamino acid residues previously developed by the Jahn group look particularly interesting since the authors report that they form, in both nonpolar organic and aqueous solvents, ordered secondary structures which are dominated by an unprecedented three-dimensional bridged triazaspiranoid-like hydrogen bond network.

This manuscript presents impressive work with the synthesis and conformational study of numerous compounds (4-, 6- and 8-mers of three families in protected or unprotected version). It's a nice combination of a number of techniques classically used to unveil preferred conformations adopted by foldamers. Unfortunately, all the data/information collected doesn't lead to a clear and strong conclusion on the preferred secondary structures of these new oligomers and particularly on the presence in solution and role of the spiro/bridged $\text{NH}(i)\cdots\text{NH}_2(i-1)\cdots\text{NR}_3(i-1)$ hydrogen bond network identified by the authors in the X-ray structure of 4RPAG-NH₂. However, I believe that this type of work is important to scientists in the field of foldamers and might influence thinking in this field but the present manuscript is not yet sufficiently refined to be published in Communications Chemistry.

My main concern is the proof of the existence of the three-dimensional bridged triazaspiranoid-like hydrogen bond network in both water and nonpolar solvents. If this particular hydrogen bond network is present in solution and contribute to the folding as said in the abstract, the evidences of its presence should be better explained and emphasized (section 2.2) - In the crystal structure of 4RPAG-NH₂, table S4 listed two hydrogen bonds for N5-H5: N5—H5···O1 and N5—H5···N4 (Why the first one is not represented in Figure 3?). Could this arrangement be comparable to unusual N—H···N Hydrogen Bond in proline (J. Am. Chem. Soc. 2014, 136, 39, 13474)? - Concerning conformational study in solution, it is written (line 177) that "The spiro/bridged hydrogen bond network is also evident in the ¹H NMR spectra of N-phenylethyl-protected hexamers 6-Fmoc and octamers 8-Boc." For the reader, this doesn't seem so obvious maybe because of a lack of explanation and graphical content to support this claim. For example, why the presence of two or three amide N-H resonances were found at $\delta \sim 7.8\text{-}8.3$ ppm, imply the presence of two or three double five-membered H-bonding $\text{NH}(i)\cdots\text{NH}_2(i-1)\cdots\text{NR}_3(i-1)$ interactions (line 178-180).

I believe that the manuscript could be strengthened by a more comprehensive demonstration of the bridged triazaspiranoid-like hydrogen bond network identified by the authors in these systems. If the manuscript is unacceptable in its present form, the study is sufficiently promising to consider a resubmission in the future.

NMR Studies (VT, titration, ROESY,) are carefully described and commented with appropriate graphical supports (Figure 4 and 5 and SI). The experimental part (SI) shows a good level of compounds characterization and provides sufficient methodological details to be able to reproduce the work.

Additional comments:

Line 27: "artificial self-assembling systems termed foldamers got in the focus of research" It seems to me that the definition of "foldamers" provided here is not accurate because the self-assembling ability is not necessary to be consider as a foldamer.

Line 31: the following review "Foldamer Catalysis" from Gellman seems more appropriate for ref 14: <https://dx.doi.org/10.1021/jacs.0c07347>

Line 40: Reference 39 doesn't really seem relevant here since this publication focuses on the reverse turn.

Line 42: "Research on foldamers concentrated only on the static relationship between given conformational constraints and hydrogen bonding between acceptors and donors along the backbone amide units in rigid structures" It would be better to say "mostly" since the structuration of certain foldamers do not rely on intramolecular backbone hydrogen-bonding network (such as peptoids for example).

The recent following review " α,β -Unsaturated γ -Peptide Foldamers"

<https://doi.org/10.1002/cplu.202100045> might be cited to complement the bibliographic part on γ -

Peptide Foldamers.

Line 63-65: The cited reference 50 from Wennemers group reports oligomers of proline bearing guanidinium groups and not ammonium substituents as discussed in the text and shown in Figure 1C. Line 80 "However, a common feature among all non-natural foldamer types is their tendency to adopt rather rigid structures in nonpolar solvents; whereas loss of folding is often observed in water." I would not be so categorical.

Line 102: (cf. Figure 1F) Figure 1F not found

The names assigned to the synthesized oligomers are counterintuitive. It will be preferable to adopt peptide nomenclature from N-terminal to C-terminal.

Figure 2: According to me, the NH for example in $\text{Pg}[\text{HN-L-Ala-(R)PAAMP}]_n\text{-OtBu}$ is not needed.

Paragraph 2.2 There is no indication about the solvents used for the crystallization of 2RPAG-Fmoc and 4RPAG-NH₂. Is it aqueous or organic solvents?

Line 175: According to me, N-aminated peptides exhibit six-membered H-bond as shown in reference 66 and 67

Figure 3, 4 and 7 show low quality resolution.

Line 294-299: I don't understand the conclusion of this paragraph. Why this implies intermolecular interactions? Could these be intramolecular interactions arising from a turn-like structure?

ECD study: Could we see a cooperative effect when increasing the sequence length (comparison from dimer to octamer) in a same series? This may support conclusion of the NMR titrations.

Figure 6: The scale and nature of the y-axis is missing on the ECD graphs.

Procedures for circular dichroism study were not found in the Supporting Information file. According Figure 6, it seems that ECD spectra were registered with a concentration in foldamer of 1 mM that seems quite high. How was this concentration chosen? Has a preliminary study at lower and higher concentrations been carried out?

An interesting point that may have been more discussed and clarified is the level of protonation of the AAMP depending on the pH. In Figure 2, the dimers 2-FmocH⁺ and 2-AcH⁺ are drawn as dicationic compounds.

Are the authors able to correlate the secondary structures proposed by the computational study (Right-handed helical structure and left-handed staircase-like structure) with experimental data obtained by NMR (VT, VC, DMSO titration). For example, more or less strong H-bonds along the backbone. The possible involvement of five-membered H-bonding $\text{NH}(i)\cdots\text{NH}_2(i-1)\cdots\text{NR}_3(i-1)$ interactions in the proposed secondary structures is not discussed. Could these H-bonds operate in these structures?

Point-by-point response to the reviewer comments

We sincerely thank the reviewers of their interest, positive view on the manuscript and constructive comments that significantly help improving the quality of the manuscript.

In the following we will address all comments in the order of their appearance in the decision letter. Moreover, we provide a highlighted manuscript version including all changes.

Reviewer 1:

We thank the reviewer for the critical and positive evaluation of the manuscript and address all points:

General comments:

No X-ray structure of larger foldamers:

We undertook multiple attempts to induce crystallization of all hexamers and octamers. While all compounds form indeed microcrystalline solids, we were not able to obtain suitable crystals for X-ray crystallographic studies. We introduce a sentence at the end of the X-ray-crystallographic section (p.6).

Energy difference prevents crystallization:

This might be a potential explanation but since the evidence is little, we prefer not to speculate on this point.

Side-by-side investigation of CD spectra from tetramer to octamer:

This is an excellent suggestion. We performed the experiments and results are commented in the manuscript at the end of the VCD and ECD section and displayed in the Supplementary information. The results clearly show no secondary structure for the dimer, dynamic features of a secondary structure for the tetramer and organized helical secondary structures in hexamers and octamers. This is documented in the SI and a sentence on p.12-13).

Mono- and dimethylated amine-containing foldamers and their structure:

This is an excellent suggestion. This is, however, a project on its own and according to the constraints of this manuscript, we decided not to pursue this for this manuscript.

Specific comments:

1. The manuscript has been condensed. The synthesis has been moved to the SI as suggested. The figure has been replaced by a structure chart of the investigated compounds (new Figure 2, p.5).
2. The synthetic scheme has been removed and replaced by just the structures (see comment 1.).
3. The full chemical structure including all H-bonds has been added to Figure 3.
4. The resolution of all figures has been improved.

Reviewer 2

We thank the reviewer for the positive evaluation and the constructive comments, which we address below.

1. The reviewer is correct that the $\text{NH}(i)\cdots\text{NH}_2(i-1)\cdots\text{NR}_3(i-1)$ hydrogen bonding is visible in the crystal structures, and we provide evidence that this motif is also present in the solution structure of the oligomers (see also reviewer 3).
2. Unfortunately, all NH_2 protons are broad and overlapped at ca. 2 ppm and thus no further information can be extracted.
3. All our MD models show a larger chemical shift of the 3,5,[7] amide NH protons, in agreement with the NMR data. The larger shifts result from their frequent $\text{NH}(i)\cdots\text{NH}_2(i-1)$ hydrogen bonds, which are now plotted in Figure 7, mirroring the pattern seen in our crystal structures. These results alone do not allow us to discriminate between the left- and right-handed patterns; however, the VCD spectra clearly indicate helical folding, predominantly right-handed. Such a structure is one of the two major folding patterns independently identified in our MD simulations. Thus, all experimental and simulation data support the pattern we propose, and although we cannot entirely dismiss the existence of other patterns, their likelihood is very low.
4. We have clarified the nature of the bonds displayed in Figure 7. Now, we clearly show which of those are NH-NH (Blue) and CO-NH (Red). The NH-NH bonds, mostly $\text{NH}(i)\cdots\text{NH}_2(i-1)$, are thoroughly commented on in the text. The CO-NH bonds are more volatile and pattern-dependent. Due to their volatility, their impact on the chemical shifts is smaller, and they are less helpful in defining the pattern, providing less structural data than their NH-NH counterparts. For this reason, their discussion is mostly omitted. We added a sentence explaining this in the text: "We focus on the $\text{NH}(i)\cdots\text{NH}_2(i-1)\cdots\text{NR}_3(i-1)$ interactions as they are the prevalent features in experiments and in our simulation models." (p.13, one but last paragraph)

Reviewer 3

We thank the reviewer for the positive evaluation and the very careful and constructive comments.

No clear and strong conclusion on preferred secondary structure in solution:

We do not fully agree to this statement. The collected evidence leads to clear conclusions. In our opinion we provide a clear line of experimental and computational results, which all converge to a preferred right-handed 18/20 helical structure. Alternatives have been evaluated. A major conclusion is that the system is more dynamic than previous reported foldamers; this naturally leads to consideration of alternative structures contributing to the global secondary structure, but this is not a weakness in our view.

Proof of the existence of the triazaspiranoid-like hydrogen bond network:

The material has been reorganized to demonstrate the existence of the triazaspiranoid hydrogen bond network. This was pursued as the reviewer asked more specifically below.

Hydrogen bonds in the crystal structure of 4RPAG-NH₂:

We are sorry for the confusion. The first N5-H5...O1 hydrogen bond is an intermolecular hydrogen bond between two molecules of the tetramer in the crystal packing, not an intramolecular hydrogen bond. The table in the SI has been checked and all purely intermolecular hydrogen bonds have been removed (Table S4, p.S54-S55).

Conformational study in solution Line 177-180:

The presence of the hydrogen bond network in the solution structures of hexamers and octamers in relation to the tetramer has now been clearly explained and will help the reader to better realize the reasoning (p.6-7).

Additional comments:

Line 27: The sentence has been changed to the accurate definition of a foldamer (p.2).

Line 31: Reference 14 was changed as suggested by the reviewer (p.2, reference section).

Line 40: Reference 39 of the original manuscript was deleted as suggested (p.2).

Line 42: "mostly" was inserted as suggested (p.2).

The suggested Review was inserted as new reference 29, original references 29-38 shifted to new references 30-39 in the revised manuscript (p.2, reference section).

Line 63-65: The reviewer is right, ammonium has been changed to guanidinium in the text (p.4) and figure 1C.

Line 80: The text was changed (p.4).

Line 102: The reviewer is right, Figure 1E is correct. The text was changed (p.4).

The names of all foldamers have been changed by naming them according to the peptide nomenclature from the N-terminal end throughout the manuscript.

Figure 2: The nomenclature has been changed throughout, see response above.

Paragraph 2.2: The solvents for crystallization have been added (p.5).

Line 175: The reviewer is right. "seven-membered" was changed to "six-membered" (p.6).

Figures 3, 4, 7: This seems to be a Word-to-pdf Windows-Mac transfer problem. All figures are high-resolution and will be properly submitted.

Line 294-299: We have clarified this conclusion, by specifically stating the two structural options and the evidence for excluding intramolecular folding based on a turn-like structure (p.10).

ECD study: This study has been performed and the cooperative effect has been clearly demonstrated (See also reviewer 1). The experimental results have been included in the SI and briefly commented in the text (p.12-13).

Figure 6: We are sorry for the missing y-axes in Figure 6. They are now included. Procedures for circular dichroism studies. This was in the SI, but under the synthetic general information. The VCD/ECD experimental setup and conditions have now been moved to the corresponding section in the SI and added to the mandatory "Methods" section (p.16-17).

Concentration of ECD study:

The spectra were collected in 0.01 cm cylindrical quartz cells at 1 mM concentration from 300 nm to 190 nm. This setup allowed to obtain maximal information about amide transitions in the far-UV spectral region for all used solvents including MeOH (cut off in 1cm cell 205nm). The use of a relatively high concentration for ECD experiments allows to compare results obtained by ECD and VCD spectroscopy. The concentration dependence of ECD spectra was tested for compound **Ac- 6GR^PA** and no spectral differences were observed. The information is in the "Methods" section (p.16-17).

Protonation of the dimer in the synthetic scheme (now in the SI): We did not investigate the dimers specifically but assume that they are fully protonated after deprotection of the t-butyl ester groups since TFA is similarly acidic as HCl.

Correlation of computed structures with NMR results and the five-membered hydrogen bond in the computations: MD and NMR data correlate well for the $\text{NH}(i)\cdots\text{NH}_2(i-1)\cdots\text{NR}_3(i-1)$ network. We are able to reproduce the large chemical shifts for the 3,5,[7] amide NH protons (now in Figure 7). The MD data also show that when these hydrogen bond networks are present, the proposed patterns appear. Unfortunately, constant pH simulations in MD are extremely complicated, especially when one wants to compute any spectra. Therefore, although we agree that these would be valuable, they go beyond the current state-of-the-art capabilities already provided in this work.

REVIEWERS' COMMENTS:

Reviewer #1 (Remarks to the Author):

The authors have done a great job updating the manuscript according to our comments and I think it can now be accepted.

Reviewer #2 (Remarks to the Author):

The improvement of the manuscript is significant: less synthesis detail, much better Figure. 2, improved resolution of Figure.6 to 7, and new labels to differentiate NH---O and NH---NH hydrogen bonds, as well as modified discussion to address reviewers' concerns.

The reviewer is still not fully convinced about the hydrogen bond network $\text{NH}(i)\text{---NH}_2(i-1)\text{---NR}_3(i-1)$ in higher order structures. The presence of the first half is evident but not the second half. The reviewer also do not agree with the author's claim (in the rebuttal letter not in the manuscript) that NH---O hydrogen bond is more volatile.

Nevertheless, the reviewer thinks the quality of the manuscript is good for publication now.

Reviewer #3 (Remarks to the Author):

The manuscript entitled "Foldamers Controlled by Functional Triamino Acids: Design and Structural Investigation of α/γ -Hybrid Peptide Oligomers" submitted for publication in Communications Chemistry by Dr Jahn and colleagues was revised taking into account the reviewers' recommendations. The authors have answered the points raised in the previous round of review and modified the manuscript accordingly.

Some remaining errors:

Many typing errors, careful proof editing is necessary

Figure 2: Gly instead of Ala in the structure of Fmoc-8ARpA and Boc-8ASpA.

p8: Fmoc-6RpAA-Fmoc

Point-by-point response to the reviewers' comments to the revision

We sincerely thank the reviewers for their interest, positive view on the manuscript and constructive comments.

In the following we will address all comments in the order of their appearance in the decision letter. Moreover, we provide a highlighted manuscript version including all changes.

Reviewer 1:

We thank the reviewer for the evaluation and recommendation for acceptance.

Reviewer 2

We thank the reviewer for the positive evaluation and the constructive comments, which we address.

We introduced a sentence in the manuscript on p.7 that the $\text{NH}_2(i-1)\cdots\text{NR}_3(i-1)$ part of the $\text{NH}(i)\cdots\text{NH}_2(i-1)\cdots\text{NR}_3(i-1)$ hydrogen bond network is evident in the solid state structure of the tetramer. The NMR chemical shift data of the pyrrolidine rings are different in hydrogen bonded and non-hydrogen-bonded state but that this remains indeed indirect evidence.

We accept the reviewer's criticism concerning the statement in the rebuttal letter. This had no consequences for the manuscript.

Reviewer 3

We thank the reviewer for the positive evaluation and the very careful and constructive comments and address here.

- The manuscript has been checked for typing errors and has been proof-read and carefully corrected.
- The peptide structures have been corrected in Figure 2 and checked throughout for correctness.
- The compound code on p.8 has been corrected.